# Anarchism Is the Only Future

**James Martel**

Department of Political Science, San Francisco State University, San Francisco, CA 94132, USA; jmartel@sfsu.edu

**Abstract:** In this paper I argue that archism, a form of political power that is ubiquitous in the world and is based on hierarchy and violence, effectively denies us a future. Archism in invested in continuing the current power dynamics. Accordingly, it projects a false sense of the future which is actually only a continuation of the present on and on forever. I look at two thinkers, Walter Benjamin and Hannah Arendt, who try to take the future back from archism (my word, not theirs). In doing so, they do not seek to determine the future but on the contrary, to allow it to actually occur in all of its infinite complexity and unpredictability, that is to say, to submit it to anarchist forms of temporality.

**Keywords:** future; anarchism; archism; Walter Benjamin; Hannah Arendt

## 1. Introduction

Anarchists have a lot of good reasons to be suspicious of the future. Not the actual future, moments in time that have yet to occur, but rather the concept of the future as such. The future, in this other, more metaphysical sense, has had a long history of supporting, not anarchism but rather archism (which, as I will explain further, is not exactly the opposite of anarchism), a system of violence and hierarchy that is so ubiquitous in the world that it does not—normally—even have a name. The future is one of those central concepts that help to keep archism viable. It is the place where ends happen, where teleologies culminate and eschatologies finally work themselves out. The future is a kind of (false) promise that, no matter how terrible things may seem in the present, there is a plan, a grand scheme which ensures that everything will work out just fine (and maybe especially well for those who are the suffering the most). The future is a guarantee and a stopgap to prevent any sense that archism is itself unjust and that it must be upended.

In this essay, I will look at the concept of the future from the position of anarchism, featuring the works of Walter Benjamin and Hannah Arendt, both anarchists in my view (the former more clearly than the latter) and both skeptical of the idea of the future as an end that resolves political questions. I will show how, for Benjamin, the past is the preferred temporal modality while for Arendt, it is the present. Yet for both thinkers, I will argue, this does not rule out an engagement with the future as well. But this engagement will be with the actual future, not a phantasm, a projection of would-be archons as a way to sustain their present power. I will ultimately argue that the archist notion of the future is actually a form of bad presentism where the present power configurations are meant to be extended indefinitely for the sake of those already in command. This effectively means that under conditions of archism, time is not meant to exist at all, not meant to bring any new threats to existing power configurations, even as it is employed as a device to make it seems as if we are plunging headlong into a fantastic and ever expanding future. Yet, for all of this, a different, anarchist, future—that is to say an actual future, an actual engagement in real time—is not only possible but I would argue, via the work of Davids Graeber and Wengrow, that it is also highly likely.

## 2. Archism and Anarchism

To begin this essay, let me say a bit more about what I mean by archism, a word, that, as already noted, is not in wide usage but which refers to a phenomenon that is nearly (but

definitely not entirely) global at this point. Archism is a form of projection of authority, the assertion of a deep, ontological basis for power that is in fact based on nothing at all. Archism is akin to what Benjamin calls "mythic violence" [1]. This is a system of control that is eternally insecure about its right to exist whereby it resorts to violence in order to reassert its power over and over again. Archism, I would say, is the name of the system and mythic violence is the means by which that system sustains itself. Archism can never be so violent as to make itself ontologically whole yet, by the same token, it cannot abstain from being violent either because, at the end of the day, violence is all that it is.

Archism can take many forms but its two principle forms are liberalism and authoritarianism (a form that includes Italian and Spanish fascism and Nazism but also pseudo-democracies like contemporary Russia as well as states that practice their authoritarianism more openly such as contemporary China). While these systems are often seen as antithetical to one another, in fact, they are two sides of the same coin, protecting capitalism through a homeostatic process of moving back and forth between these two instantiations [2]. When capitalism is relatively unchallenged, a liberal form of archism prevails, convincing many through its doctrines of individual achievement that those who are lower in the social and economic hierarchies have a fair chance at success (they do not). When, however, the elites sponge up so many resources (as is currently the case) that the imbalance threatens the market principle in a fundamental way, the more openly violent and authoritarian face of archism emerges to terrify its enemies into submission. Eventually, when any opposition to capitalism is sufficiently suppressed, the liberal form, which is more stable and less costly, returns.

Both liberalism and authoritarianism, when understood as manifestations of archism, can be seen as forms of bad immanentism wherein the desires of the archons are associated with the desires of some great externality. This externality used to be God but it has become secularized into various forms such as reason or nature, the "will of the people" or tradition. The faux transcendentalism of archism is at its core a physical manifestation of a metaphysics of hierarchy and control. An externality is relied upon but only to bolster and justify intense political, economic and social inequality which lies at the center of archism (its very purpose).

Yet, for all of its terrifying power, archism is uniquely vulnerable because it relies on instilling a belief in its ubiquity and inevitability as a way to sustain itself. The liberal variety of archism readily admits to many of its faults—its production of economic and social inequality, its racism and misogyny—but claims that "there is no alternative," (the alternative being either authoritarianism which, especially in its fascist mode, often serves as a liberal boogeyman or a depiction of communism as being essentially the same thing). The authoritarian variety of archism admits nothing but, as already noted, at least in its more openly violent forms, tends to be unstable and short lived. This is why giving this system a name of its own, archism, is so critical. Even to point out that it is something and not just everything, reduces and challenges archism's power, its faux metaphysics. If archism is not the universal and not the future, then it is in fact nothing. Indeed, it has always been nothing, a purely parasitical power feeding off the subjects that it holds in thrall. Yet, so long as we give it credence, see it as our end, our destiny and our future, that hold remains.

Anarchism is something entirely different. Normally, we ascribe the word anarchism to a European-based tradition, focusing on the work of figures like Bakunin, Kropotkin and Goldman. Yet, I would argue that when we introduce a word like archism, the term anarchism itself necessarily becomes something much bigger. Insofar as it is "not archism," anarchism is every other practice in the world, every other way for human beings to arrange themselves politically, socially and economically. In their recent book *The Dawn of Everything*, David Graeber and David Wengrow (henceforth the two Davids) show that, contrary to the way that history and prehistory are usually depicted, the history of the world has not led inevitably to the modern archist form (they use neither the word archist nor anarchist in their text so this is a gloss that I am making on their argument) [3]. Rather

than leading to Western models of sovereignty, the two Davids show that in fact, when left to their own devices, human beings tend toward far more egalitarian and relatively unstructured forms of sociality. They document countless cases wherein a community that has been read as a precursor to archism is shown to be nothing of the kind.

To give just one of many, many examples that the two Davids offer, they speak of how Minoan Crete has long been seen as a key example of early "state formation" ([3], p. 434). The ruins of the "Palace" at Knossos has been read by mapping an assumption of monarchy and hierarchy onto the stones and mosaics that remain there. Yet, they note that, in fact, "there is simply no clear evidence of monarchy on Minoan Crete" ([3], p. 434). They further note that the artwork depicts a clear matriarchy, contrary to many more patriarchal themes found in other cultures' artistic displays. They also show that there were almost no references to warfare in the Minoan depictions but rather they mainly featured moments of play and "common creature comforts" ([3], p. 437). Perhaps most telling of all is that the room that has long been called a "throne room" seems more suited to a council with two sets of benches facing each other and a ritual bath nearby ([3], p. 437). The assumption that this must be a throne room (or even that it takes place within a "palace") indicates the way that the imagination of contemporary archeologists has been influenced by archist teleologies where there is only one way to organize a community (by archist means). Such thinking hides the fact that human life has been far more diverse and complex in the way that it approaches such questions. The two Davids argue that throughout most of human history, very often when a kind of top-down state-like formation was attempted by some elite, the response was to have people simply leave or to overthrow their would-be archons by revolt or revolution.

In this way, I think it would be a mistake to say that archism is the "opposite" of anarchism or that the terms archism/anarchism function as a binarism. Archism is one, very bad, very violent method of organizing political and economic power. Anarchism is everything else. Arguably, archism contains all of the worst elements that are found in many other societies. The two Davids note the use of slavery, of warrior castes, or other kinds of violence in some of the examples that they look at, but none to the same extent that archism features. Without the kinds of psychic anxieties that mark archism (due to its false projections of authority to mask its illegitimacy), these societies, in all of their variety, do not need to resort to endless violence as such. This does not guarantee that everything is perfect in anarchist societies; only archism promises perfection and that promise is itself a lie. But it does mean that non-archist societies are not driven by the same demand and anxiety about their right to exist than archism is itself.

This analysis permits me to make a somewhat optimistic claim about the future as a way to segue towards that topic more generally. Here, I am not talking about the "future" that archism uses to promote a kind of timelessness where archism, capitalism and the modern state will exist forever so that it is in fact not a future at all (as I will explain further shortly), but the actual human-produced future, one that remains utterly unscripted and as yet unknown.

If the two Davids are correct in their analysis of our past, then it stands to reason to say that there is at least a very strong possibility that, once we are done with the nightmare of archism, we will revert to something far more equitable and just. Because archism is so violent, so racist, sexist, queerphobic and hostile to human life as a whole, it is very hard to maintain, requiring a lot of buy-in from some people and at least partial acquiescence from most others. Assuming that when this all falls apart and the world is still there or also that archism has not killed us all off with fascist and authoritarian wars or global and environmental catastrophe (very big ifs, I know), it seems likely that the human race will revert to the non-archist forms of life that it has practiced for far longer. Archism, for all its fury and spectacle, is, in the span of human history, just a blip, albeit a horrific and unsurpassably bloody one and it may be that the future of humanity will look back on our time and wonder how it was possible that it even existed for as long as it did (or even existed in the first place).

### 3. Benjamin: Disenchanting the Future

Having better established the distinction between archism and anarchism, let me now turn to the question of the future more generally and show how two thinkers, Benjamin and Arendt, think about it. Both thinkers offer us resources from within the maw of archist power with which to struggle against the way that archism has colonized the future as a concept, making it serve their nefarious purposes. Above all, both of these thinkers show us how what we normally call "the future" is nothing of the sort. Rather it is a fabrication, one that is superimposed over all of human temporality to deny a space within which anything could ever be different than it is right now.

Beginning with Benjamin, we see that in his final essay "On the Concept of History," written at the very end of his life, Benjamin engaged directly with the question of the future, largely by seeming to reject this concept out of hand. As I will show, this is not the case. Yet, in order to get at Benjamin's understanding of the future, it is first necessary to explain what he thought about the past and present and how it relates to this other temporality. In "On the Concept of History," Benjamin makes a strong distinction between what he calls "historicism" and "historical materialism." Roughly speaking, these terms may correspond with what I've been calling archism and anarchism (of course, historical materialism is associated with communism but one of my contentions is that Benjamin's version of communism is deeply anarchistic) [4–7]. He tells us that

> The historical materialist cannot do without the notion of a present which is not a transition, but in which time takes a stand [*einsteht*] and has come to a standstill. For this notion defines the very present in which he himself is writing history. Historicism offers the "eternal" image of the past; historical materialism supplies a unique experience with the past" ([8], p. 369).

Here, we see that for Benjamin, the historical materialist understands that time itself has been brought to a "standstill." The present is, as Benjamin tells us, a kind of endlessness; as already noted, it is the perpetuation of the same power system that superimposes itself over the past, present and, of course (and this is the entire point), the future as well.

Given this, the historicist, that is to say, the agent of archism (again this is my term, not Benjamin's) sees the present as a kind of "eternal image" whereas the historical materialist sees a "unique experience of the past." This distinction is worth parsing a bit further. The historicist presents an eternal image of the past because in every temporal direction that they look, they superimpose one continuous (and false) present, something that never changes. Arguably, you could say that this kind of eternal presentism being extended to the past may help explain the situation in archeology and anthropology that the two Davids confront in their book. This idea exemplifies the notion that the past is always "leading up to" the present, that the present is in some ways foreordained. By the same token, the future is similarly implicated in the decision to search for kings, palaces and throne rooms in every grand ruin from the ancient world. Here, the idea is that human beings have always existed hierarchically and will always exist hierarchically.

So far so good. We can readily see the problems with historicism and its view of time (essentially the erasure of history as such). But what does it mean for the historical materialist to have a "unique experience of the past?" Here, we see how the materialism of that latter term actually plays itself out. For Benjamin, the past is very different from the future in that it has actually happened and remnants of it still exist in our own time (that is to say that they exist materially, whether as indicators of some past event or even memories or stories of those times insofar as these things are preserved in images, texts and words). Benjamin tells us:

> Articulating the past historically does not mean recognizing it "the way it really was." It means appropriating a memory as it flashes up in a moment of danger. Historical materialism wishes to hold fast that image of the past which unexpectedly appears to the historical subject in a moment of danger. The danger threatens both

the content of the tradition and those who inherit it. For both, it is one and the same thing: the danger of becoming a tool of the ruling classes ([8], p. 391).

Rather than fetishizing the past (which would only mean to accept the eternal image cast upon it by the archons), Benjamin seeks to have an object or event or moment from the past come into contact with the present as a way to destabilize both the past and the present by interfering with one another. The clash between one part of an "eternal now" and another, being so visibly and distinctly different, disrupts and disallows the sense of a timeless continuum for Benjamin (he also tells us that "*even the dead* will not be safe from the enemy if he is victorious," indicating the past too is vulnerable to the stultifying effects of archism and is just as much in need of liberation from false temporalities as the present and the future) ([8], p. 391).

In this way, the past serves as a basis for the overcoming of the effort to get rid of time altogether, or more accurately, it contests the way that archism turns all of time into one solid thing, what he calls "homogeneous, empty time" ([8], p. 395). Benjamin specifically admonishes us that a society that looks only to the future is doomed to repeat the failures of the past, stating that "this indoctrination made the working class forget both its hatred and its spirit of sacrifice for both are nourished by the image of enslaved ancestors rather than by the ideal of liberated grandchildren" ([8], p. 394). In other words, so long as would be revolutionaries look to the future, they are condemned to seeing only the empty anti-temporality of archism itself. For Benjamin, it is not the future but the past which offers a material form from which to contest the phantasms that archism uses to overwrite reality as a whole.

In concluding this essay, Benjamin tells us famously that:

We know that the Jews were prohibited from inquiring into the future: the Torah and the prayers instructed them in remembrance. This disenchanted the future, which holds sway over all those who turn to soothsaying for enlightenment. This does not imply, however, that for the Jews the future became homogeneous, empty time. For every second was the small gateway through which the Messiah might enter ([8], p. 397).

In speaking of "disenchanting the future," Benjamin is allowing that the future need not be the "empty homogeneous time" that archism (or in his view, mythic violence) would reduce it to being. The fact that every second is "the small gateway through which the Messiah might enter" offers that time cannot be permanently and entirely determined by the false transcendentalism of myth and reaction. It also suggests that we can unmake the hierarchies of time itself (so as to deny that the future is better than the present or the past for example). Every second of time is potentially a messianic rupturing of the power of archism and this rupturing cannot be prevented whether we are looking forward or backward through time.

Although an allusion to messianism might seem to suggest the very form of future salvation that Benjamin is more generally opposed to, this is not the case. In his "Critique of Violence," Benjamin counters the power of mythic violence with an answering divine violence. Divine violence (Godly violence might be a better translation) comes from an actual deity and so it does not share the anxiety of mythic violence about its right to exist (a right that it must continually assert through bloodshed). Divine violence is a form of violence indeed but, as Benjamin explains it, only insofar as it seeks to unmake the lies and projection of mythic violence. Unlike more traditional notions of messianism, for Benjamin, the advent of divine violence creates no new truths of its own; it is a wholly negative power. In this way it does not so much occupy the future as it makes a different and real future possible in the first place.

One way to understand the difference between mythic and divine violence in Benjamin is to turn to Machiavelli's invocation of Numa, the second king of Rome. Numa feared that the Romans were unlawful so he lied and said that a goddess had given him sacred tablets of law and that the Romans had to obey them least they die ([9], p. 147). This ruse worked

and so Rome became, you could say, an early adapter to mythic violence, where the desires of the archon (in this case Numa) is projected outward (in this case onto the goddess Egeria, as Livy tells it) and returns in the form of a law that cannot be denied. But imagine if the real Egeria actually showed up, took the tablets that had been falsely attributed to her name and broke them into a million pieces, leaving the scene without ever uttering a word. That would be an instance of divine violence, a messianic intervention that counters the lies of mythic violence without instantiating any new beliefs and, in so doing, leaving a space for human beings to decide for themselves what to believe, what laws to follow (or not). In this way, I would say that whereas mythic violence is, as already noted, entirely bound up with archism, divine violence is not the source of, but rather makes possible, the human practice of anarchism.

In light of this, when Benjamin seeks to counter the lies that archism tells about the future, he turns to what he calls a "*weak* messianic power" to resist the lies of mythic violence that belongs to each generation of human beings ([8], p. 390). As can be seen by his use of the term "*weak*", this power is not itself transcendent or all powerful. For Benjamin, human beings do not directly yield divine violence but they can benefit from its disruption of myth. The upshot of this claim is that in the face of divine violence, human beings have a power of their own to enact their own messianic self-redemption. This power is weak because it is derived from divine violence but it is not in fact actually powerless.

Human beings can operate this weak messianic power by engaging in the task of historical materialism; taking advantage of the disruption of archist lies that divine violence produces, they juxtapose real elements of the past with the present in order to break the monopoly on reality that is presented by archist phantasm and projection. In doing so, it becomes possible to reclaim (i.e., "disenchant") the future itself. Here, the false uniformity that archism seeks to impose on the future gives way to an anarchized future where human agency and diversity of outcomes can be fully expressed.

It should be clear from this that Benjamin's concept of historical materialism is not precisely the same as that used by more conventional Marxists in that it has a strong theistic element. Yet, in effect, that messianic element (the disruptive power of divine violence itself) only returns us to the actual material world and so, in the end, it resembles historical materialism in its more conventional sense minus any hidden teleologies or other archist elements that might come along with the more secular version.

Thinking in this way, we can see that anarchism is itself a form of disenchantment. Against all the magic and mystery of archism, of false projections onto gods and externalities that do not exist, anarchism offers a much stronger connection to reality itself via the processes of historical materialism. This is not "reality" as in an unproblematic and obvious set of objects that are self-explanatory (that passel of lies is a trademark of archism) but rather reality as in the effects of engaging with the materiality of time itself, its remnants, its traces and its possibilities for opening up the future.

For Benjamin, engaging in historical materialism uncovers the anarchist life that political communities are always engaging in even as they are also subjects to archist power. As Benjamin tells us in the "Critique of Violence": "Mythic violence is blood-violence over mere life for the sake of itself; divine violence is pure violence over all of life for the sake of the living" ([1], pp. 57–58). It is this concept of "the living," the idea that we engage in mutual and collective practices, even as we are also reduced to mere life under conditions of archism, that Benjamin seeks to recoup. This opens up a strange duality where we both are subjects of archism and agents of anarchism at the same time. As creatures with "mere life," we do not have a future, only the stasis of an eternal now (in a false, only pseudo-transcendent sense). Yet at the exact same time, as members of the "living," we do have a future, one that it is up to us to both recognize and forge. Were it not for this duality there would be no chance at all for archism's grip on us to ever fail but the good news is that insofar as we are always practicing a form of daily anarchism, even if we do not necessarily recognize it as such, we do not have to reinvent the world. The world that

the two Davids describe is still here and people have not changed in any fundamental way. This is why we can be said to have a future at all (and an anarchist one at that).

## 4. Arendt and the Future of the Present

For her own part, Arendt shares Benjamin's concern that the idea of future is readily overwritten with a sense of predestination or teleology. She too wants to keep the future unknown, unexpected and free from archist interference (once again, not her word either and in her case, I suspect that she might have objected to its usage as she is interested—in a positive way—in the Greek verb *archein* from which it is derived, meaning both to begin and to rule). Although Arendt's anarchism is less full-throated than Benjamin's—a bit of an understatement—she too is interested above all in untrammeled collective power and for her as well there is a critical temporal dimension to this [10–12].

Arendt begins *The Human Condition* by arguing that technological solutions to human problems such as leaving the planet to escape its devastation or developing superior forms of being (like AI, although she could not have known that development in advance) are problematic in that they threaten to "cut...the last tie through which even man belongs among the children of nature" ([13], p. 2). She goes on to say that given the possibility of "this future man... it could be that we, who are earth-bound creatures and have begun to act as though we were dwellers of the universe, will forever be unable to understand, that is, to think and speak about the things which nevertheless we are able to do" ([13], pp. 2–3).

In other words, the very idea of the future as one that is controlled and determined by scientists and experts is one which will be denuded of politics which, for Arendt, is our true and ontological purpose. As she goes on to say (as ever, using the term "man" to refer to all people) "Men in the plural, this is, men in so far as they live and move and act in this world, can experience meaningfulness only because they can talk with and make sense to each other and to themselves" ([13], p 4). If the future is only going to be a place where human beings seek a "rebellion against human existence" ([13], p. 2), then it is going to be effectively nothing at all, akin to Benjamin's "empty homogenous time." As she says later in the book "a life without speech and without action....is literally dead to the world; it has ceased to be a human life because it is no longer lived among men" ([13], p. 176).

Above all, for Arendt, the key human capacity is to generate the new and the unexpected. This is what distinguishes us, in her view, from animals and which is the source of action, the highest human capacity which is also the basis for politics. As she tells us:

> The fact that man is capable of action means that the unexpected can be expected from him, that he is able to perform what is infinitely improbable. And this... is possible only because each man is unique, so that with each birth something uniquely new comes into the world ([12], p. 178).

This discussion connects with Benjamin's own concerns insofar as Arendt too fears an overweening power by the state or other sovereign forces which interfere with and actually replace this kind of permanent unknowability and surprise with something that is very knowable and predictable indeed. Here, once again, the future as such is jeopardized by overwriting and superseding doctrines that seek to reduce human plurality to one great (and ultimately totalitarian) unity.

Where Benjamin looks to the past as a source of resistance to this false futurity, Arendt looks to the present, to the moment of indeterminacy itself and the way that human actors are a surprise to themselves and to others when they act in concert with others. It is this radical presentism that offers Arendt her greatest buttress to any attempt to overwrite or supersede human agency.

For Arendt, a key vulnerability for human beings, what makes them susceptible to what I have been calling archism, is their understandable fear of the future (in another text, she calls this the "abyss of freedom") precisely because it is unknown ([14], p. 195). Because, as she tells us, when we act, we cannot anticipate nor control the consequences of that action, this leads to a fear of our own responsibility and the irreversibility of what we do. Yet, rather than have some archon step into that gap and determine for us what we

can and cannot do and say (which is anathema for Arendt), she tells us that: "The remedy for unpredictability, for the chaotic uncertainty of the future, is contained in the faculty to make and keep promises" ([13], p. 244).

By way of example, in *On Revolution*, Arendt speaks about the Mayflower Compact that was signed by Puritans fleeing from Britain, setting up a new social order from scratch (an order built on the extermination and removal of an entirely different social order, which is not something that Arendt engages with at all). Although the signatories of the Mayflower Compact were largely strangers to one another, she tells us:

> The really astounding fact…is that their obvious fear of one another was accompanied by the no less obvious confidence that they had in their own power, granted and confirmed by no one and as yet unsupported by any means of violence, to combine themselves into a 'civil Body Politick,' which [was] held together solely by the strength of mutual promise ([15], p. 167).

Speaking more generally in *The Human Condition* about the power of promising to combat the fears that opening up a radically new future may bring, she tells us that:

> The function of the faculty of promising is …the only alternative to a mastery which relies on the domination of one's self and rule over others; it corresponds exactly to the existence of a freedom which was given under the condition of non-sovereignty. The danger and the advantage inherent in all bodies politic that rely on rule and sovereignty, leave the unpredictability of human affairs and the unreliability of men as they are, using them merely as the medium, as it were, into which certain guideposts of reliability are erected. The moment promises lose their character as isolated islands of certainty in an ocean of uncertainty, that is when this faculty is misused to cover the whole ground of the future and to map out a path secured in all directions, they lose their binding power and the whole enterprise becomes self-defeating ([13], p. 244).

I quoted this last passage at length because it offers a concise illumination of Arendt's understanding of how the future can be recouped. First of all, we see how for Arendt, just as for Benjamin, the "whole ground of the future" can be covered and secured, eliminating human freedom and the possibility of action in the process.

The faculty of promising is Arendt's solution to this. It is what gets us to think beyond our own limitations and our fears of taking responsibility on our own. Promising projects us as actors into an as yet unknown future offering "islands of predictability" and "certain guidelines of reliability." These are not guarantees; once again, a guarantee is a hallmark of archist politics and the whole point of both Arendt and Benjamin's understanding is that we must accept the fact that nothing is assured in order to celebrate and expand on human freedom.

Rather than turn to externalities and fake assurances from higher powers (the veritable fox guarding the hen house), Arendt would have us turn to one another, a highly anarchist concept if there is one. We cannot guarantee but we can promise that, whatever happens, we will have one another's back. The "islands of predictability" that this suggests makes the future itself palatable and thus allows us, as with Benjamin, to keep it undetermined so that it does not simply replicate the present.

Arendt says further of this that such an action "derives from the capacity to dispose of the future as though it were the present," that is, it serves to create a best of both worlds situation in which the future is both like the present, seemingly contained with the envelope of collective action and public commitment, even as it is also in some sense wild and unfettered ([13], p. 245). This offers a vision of anarchist time that speaks both to the way that anarchism is both highly organized even as it is also not bound by any one set of rules or expectations.

## 5. Conclusions: The Future of the Future

Having examined the ideas of the future of both Benjamin and Arendt, we can see that anarchism (and here I mean it both in its conventional sense as well as in its expanded sense as "not archism") is not only consistent with the future but is also perhaps the only possible way for there to be a future at all. Against the archist desire to control, and actually eliminate, time (Benjamin tells us that that the advent of capitalism achieves the "temporality of hell," an unending sameness), human beings have the power and the capacity to make the future something real ([16], p. 66). For Benjamin, this is, once again, a "*weak* messianic power," a divine power that has been delegated to human beings themselves for their own self-salvation. Here, the coming of the messiah and the coming of the revolution are one and the same event. For Arendt, the power to create a future together is an ontological certainty given to us by the fact of our birth into a world already populated by other people. For both thinkers, this is a power that can never be taken away from us but it can be overwritten, superseded by archist lies and projections.

At this point, I want to reiterate my rather optimistic conjecture that, at the end of the day, the long nightmare of archism will, indeed must, come to an end. Drawing once again upon the two David's *The Dawn of Everything*, we can think of the immense diversity of human experience and collective decision making as a kind of laboratory where the same experiment is conducted over and over again. As a community forms, it must ask itself some basic questions. How will we govern ourselves? Will anyone be in charge? How will we deal with questions of the distribution of goods and equity? What will be the status of women in relation to men (or vice versa)? How will we deal with differences more generally? In the nearly infinite variety of answers to these questions that the two Davids explore in human history and prehistory, we see that, only very rarely have the answers come to even approximate those of archism.

The two Davids furnish examples that show that, in some ways, this is no accident. Societies have often worked very hard to avoid turning into a form of archism, not because it is teleological destiny for human beings but rather because there are always going to be bad actors who want to amass more for themselves. Or perhaps it is not even a matter of bad actors but rather of changes in society producing forms of inequality that then become standardized, self-re-enforcing and needing defending and so forth. The two Davids discuss, for example, the *ayllu* system that began in the Incan empire and which perseveres in the Andean highlands among indigenous communities to this day. The *ayllu*, a series of groupings of family that work together towards common economic and political goals, was the basis for Incan society. They formed a kind of bureaucracy whose function it was precisely to avoid any one group amassing more than other groups. The two Davids write:

> *Ayllu* too were based on a strong principle of equality; their members literally wore uniforms, with each valley having its own traditional designs of cloth. One of the *ayllu's* main functions was to redistribute agricultural land as families grew larger or smaller, to ensure none grew richer than any other ([3], p. 642).

Here, we see evidence that preindustrial societies were well aware of the dangers posed by inequitable distributions of wealth and other such matters. They considered it their business to work together to ensure that they remained free and undominated (Mariátegui, in his own writings on the *ayllu* acknowledges that they operated under the aegis of an empire but he says that, in day-to-day life, the empire had virtually no power or effect on these communities) [17].

Given the prevalence of non-archism in human history, and given the huge costs of maintaining archism itself, I do think that a future without archism is not only plausible but likely. I say this because I truly think that archism as such is unsustainable. Either we all die (that would probably be archism's choice) or we live in other ways.

The key thing to keep in mind about the future that we learn from Benjamin and Arendt is that it does not (yet) exist in any way. It is radically open-ended and undetermined. Archism's attempt to make the future accessible and controllable is an extension of its own

larger insecurity about its right to exist (and even about its existence at all). As Arendt shows, one of the ways that archism does this is by exploiting our own anxiety about responsibility for our actions and the "abyss of freedom" but she also shows that we can beat archism at its own game by offering, not guarantees, but promises, as in "I promise that if something bad happens, I will be there for you". That kind of assurance mimics the way that archism papers over the future yet here; instead of actually papering over and superseding, this promising allows the future as such to be, to exist at all.

The future as such is inherently anarchist. It is a million possibilities not just one. Archism would seek to block and control that possibility but it cannot prevent it. The future, just like human diversity more generally, is something that archism cannot undo. It sits like a parasite atop this anarchist ferment. It can kill its host but it cannot change it. It draws its own lifeblood precisely from the variety and multitude of human agency. The anarchist core of human life, what Benjamin calls "the living," is a constant but unlike the archist constant, it seeks not to stamp out but to promote human flourishing and the infinite ways that we can be and act together in real time, in an actual future.

**Funding:** This research received no external funding.

**Institutional Review Board Statement:** Not applicable.

**Informed Consent Statement:** Not applicable.

**Data Availability Statement:** Not applicable.

**Conflicts of Interest:** The author declares no conflict of interest.

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
