# Peer review of "Anarchism Is the Only Future"

_philosophies, doi:10.3390/philosophies8060113_

Round 1

Reviewer 1 Report

Comments and Suggestions for Authors

This essay challenges readers to take seriously anarchist alternatives to the violent and exploitative present society by starkly warning us that state-centric (archist) power structures are likely to kill us all:  "Either we all die...or we live in other ways."(p. 8) The other ways become visible by pivoting to the kinds of order embodied by societies not organized by top-down hierarchies and systematic violence.

Most of the essay is taken up with contrasting the views of history and temporality of Walter Benjamin and Hannah Arendt with those of archism. The author explores Benjaminʻs and Arendtʻs stress on open-ended futures, on each present as a potentially new begining, as opposed to the more standard timelines of archist societies in which the future and present are expressions of the logic of the past. The possibility of fresh starts, rather than continuous re-treading of a single past, is the grounds of a non-archist turn.

This argument is compelling, but for people who arenÊ»t scholars of Walter Benjain, it contains some puzzles. The author is persuasive regarding the limitations of hegemonic time, but less clear on BenjaminÊ»s alternatives. What, exactly, is “divine violence”? Why is it better than “mythic violence”?  Why should we agree to call this alternative temporality “historical materialism,” when in many hands, historical materialism has been constituted as much more like the problematic “historicism” in which future stages of history just naturally emerge from the logic of past stages?

The discussion of Arendt leaves fewer puzzles. The author connects ArendtÊ»s ideas about how we live in an open future to her commitment to the power of speech and action as a “glue” that holds us together, including the ability to make promises. I like this discussion very much, but it is quite brief. I long for some examples from ArendtÊ»s work to flush out the sketch of ideas.

Author Response

Thank you for your review. I added more about the difference between mystic and divine violence and also included an example from Arendt that illustrates what she is talking about about the future.

Reviewer 2 Report

Comments and Suggestions for Authors

The author needs to 

1. cite the scholarship interpreting Walter Benjamin's ideas as anarchist

 and

2. engage with and contextualise their claims concerning Benjamin's concepts referencing other scholarly treatments of Benjamin's ideas, and specifically his conception of history, as anarchist.

3. The author needs to disgard cavelier references to "Fascism" as the antithesis to "Liberalism" and substitute the broader and generic term "authoritarianism" for "Fascism" throughout the article.  Fascism is a distinct political movement with specific features, none of which the author evidences any grasp of. Citing the "two Davids" Dawn of Everything without providing a page reference (see reference ii) underlines this imperative.  If the subtitution is made, it will save the author the need to actually discuss what Fascism is.  If, on the other hand, the author insists on referencing Fascism as the polar opposite of Liberalism along the Archism continuum, then I they need to consult the appropriate secondary literature and cite it.  

5. The author needs to provide a scholarly definition of Liberalism, citing the appropriate secondary sources.

6. The author needs to expand on their interpretation of Arendt's ideas as anarchist referencing the relevant secondary literature.  I recommend, in particular, that they consult Jimmy Klausen's "The Abivalent Anarchism of Hannah Arendt" in the anthology, How Not to Be Governed.

Author Response

Thank you for your review. I made changes that you suggested including switching from the term fascist to authoritarian and doing more citing in the literature about the anarchism both of Benjamin and Arendt.

Reviewer 3 Report

Comments and Suggestions for Authors

This was a clearly written and well-argued paper about the limitations of archism in its attempt to determine the future. Instead, the author develops an original claim for anarchism as a way of thinking that leaves the future open to infinite possibilities. Here, anarchism is understood not so much as a political ideology but as a way of living in the world and affirming the contingency and plurality of life; whereas archism is the attempt to impose a totalizing order (of sovereignty, law, capitalism etc) on the world, but which is actually parasitic on life. Here the author draws on anarchistically-inclined thinkers - Benjamin and Arendt - to show that anarchism is actually the 'normal' way of living, not some ultimate goal of revolutionary politics. The paper is original, engaging and persuasive, and makes an important contribution to anarchist political philosophy. I would recommend publication.

Author Response

Thank you so much for your kind and supportive review.